# An Overview of PRR- and NLR-Mediated Immunities: Conserved Signaling Components across the Plant Kingdom That Communicate Both Pathways

**DOI:** 10.3390/ijms232112974

**Published:** 2022-10-26

**Authors:** Candy Yuriria Ramírez-Zavaleta, Laura Jeannette García-Barrera, Lizette Liliana Rodríguez-Verástegui, Daniela Arrieta-Flores, Josefat Gregorio-Jorge

**Affiliations:** 1Programa Académico de Ingeniería en Biotecnología—Cuerpo Académico Procesos Biotecnológicos, Universidad Politécnica de Tlaxcala, Av. Universidad Politécnica 1, Tepeyanco 90180, Mexico; 2Instituto de Biotecnología y Ecología Aplicada (INBIOTECA), Universidad Veracruzana, Av. de las Culturas, Veracruzanas No. 101, Xalapa 91090, Mexico; 3Centro de Investigación en Biotecnología Aplicada, Instituto Politécnico Nacional, Carretera Estatal Santa Inés Tecuexcomac-Tepetitla Km.1.5, Santa Inés-Tecuexcomac-Tepetitla 90700, Mexico; 4Departamento de Biotecnología, Universidad Autónoma Metropolitana, Iztapalapa, Ciudad de México 09310, Mexico; 5Consejo Nacional de Ciencia y Tecnología—Comisión Nacional del Agua, Av. Insurgentes Sur 1582, Col. Crédito Constructor, Del. Benito Juárez, Ciudad de México 03940, Mexico

**Keywords:** plant immunity, pathogen, disease, receptors, signaling, evolution, network

## Abstract

Cell-surface-localized pattern recognition receptors (PRRs) and intracellular nucleotide-binding domain and leucine-rich repeat receptors (NLRs) are plant immune proteins that trigger an orchestrated downstream signaling in response to molecules of microbial origin or host plant origin. Historically, PRRs have been associated with pattern-triggered immunity (PTI), whereas NLRs have been involved with effector-triggered immunity (ETI). However, recent studies reveal that such binary distinction is far from being applicable to the real world. Although the perception of plant pathogens and the final mounting response are achieved by different means, central hubs involved in signaling are shared between PTI and ETI, blurring the zig-zag model of plant immunity. In this review, we not only summarize our current understanding of PRR- and NLR-mediated immunities in plants, but also highlight those signaling components that are evolutionarily conserved across the plant kingdom. Altogether, we attempt to offer an overview of how plants mediate and integrate the induction of the defense responses that comprise PTI and ETI, emphasizing the need for more evolutionary molecular plant–microbe interactions (EvoMPMI) studies that will pave the way to a better understanding of the emergence of the core molecular machinery involved in the so-called evolutionary arms race between plants and microbes.

## 1. Plant Immunity at a Glance

Since their emergence around 500 million years ago (MYA), land plants have been exposed to constantly changing conditions [1]. Besides coping with fluctuating environmental conditions and adjusting their physiology, plants are also challenged by microbes (bacteria, fungi, viruses, and protists) living in their surroundings, some of them beneficial and some pathogenic [1,2]. To navigate this microbial world, the plant must recognize and promote healthy interactions but simultaneously reduce those interactions that could cause disease [3]. Unlike beneficial microorganisms, pathogens cause damage to their hosts through various means, including the production of toxins, cell-wall-degrading enzymes, or virulence proteins [4]. Accordingly, plants have evolved mechanisms of perception and signaling for mounting a proper response depending on the nature of the interaction [5]. This three-layered system begins with the detection of molecules that comprise various microbe external features such as peptides, chito-oligosaccharides, or lipo-chitooligosaccharides, collectively known as microbe-associated molecular patterns (MAMPs) (Figure 1). The recognition of MAMPs is carried out by cell-surface-localized pattern recognition receptors (PRRs). PRRs are either receptor-like kinases (RLKs) or receptor-like proteins (RLPs) [6]. In any case, both kinds of receptors perceive MAMPs via a repertoire of extracellular domains and require the assistance of co-receptors to transduce the signal, resulting in calcium (Ca^2+^) influx, calcium-dependent kinases (CDPKs) activation, reactive oxygen species (ROS) burst, mitogen-activated protein kinases (MAPKs) activation, hormone signaling, and transcriptional reprogramming (Figure 1), all aimed at restricting the pathogen invasion [7]. Such a first response of the host is known as pattern-triggered immunity (PTI), which is evaded or suppressed by pathogens through secreted effector molecules (Figure 1). Thus, if the pathogen overcomes PTI through effectors, the so-called effector-triggered immunity (ETI) is activated [8]. ETI is a more specific response initiated by the recognition of effectors through intracellular nucleotide-binding domain and leucine-rich repeat (NB-LRR) receptors (NLRs) [9]. Like PRRs, NLRs usually detect effectors with the help of co-receptors, leading to the activation of downstream responses such as massive production of ROS, Ca^2+^ influx, activation of MAPKs, synthesis of defense hormones, defense gene expression, and programmed cell death (Figure 1) [8,10]. In summary, pathogen recognition by PRRs and NLRs results in downregulation of growth mediated by phytohormones and upregulation of defense-related genes. Failure of the host to detect and mount the proper response causes the appearance of disease, known as effector-triggered susceptibility (ETS).

The current and widely accepted “zig-zag-zig” model of plant immunity incorporates the interaction between PTI, ETI, and ETS [8]. Although PTI and ETI involve different activation mechanisms, increasing evidence suggests that they interact through common downstream signaling components, blurring the distinction between the two plant immunity branches [11]. Moreover, such convergence seems to extend in the third layer of plant–pathogen interaction (defense-action layer) since many downstream responses of PTI and ETI are similar, albeit with distinct temporal patterns and intensities. Therefore, besides offering a general picture of plant immunity mechanisms, we focus on those converging points between PTI and ETI.

## 2. Deconvoluting the PRR-Mediated Immunity: Perception, Activation, and Early Signaling Mechanisms

In the plant’s first layer of defense, perception of plant pathogens depends on the specific interactions between host receptors and the pathogen elicitors known as MAMPs. At the molecular level, MAMPs are recognized directly or indirectly by transmembrane PRRs classified as RLKs and RLPs [12]. Of note, RLKs form a monophyletic gene family and their origin predates the divergence of plants and animals [13,14], whereas RLPs are less conserved [15,16]. Specifically, plant RLKs belong to the RLK/Pelle family because these are closely related to Drosophila Pelle proteins and animal cytoplasmic kinases [14,17]. Besides their involvement in immune processes, RLKs also regulate plant symbiosis alongside growth and development [18,19,20]. Due to gene expansion and diversification, RLKs have followed a complex evolutionary history during plant evolution. Still, one thing is clear, the number and diversity of RLKs have expanded since the appearance of Streptophytes onwards, particularly more pronounced in land plants (Bryophytes and Tracheophytes) [21,22,23].

Structurally, RLKs and RLPs are composed of an extracellular ligand binding domain (ECD), a transmembrane domain, and in the case of RLKs, a cytoplasmic kinase domain that RLPs lack. Instead, RLPs possess a short cytoplasmic region (Figure 2A). Since ECDs mediate the recognition of MAMPs, RLKs and RLPs can be classified into sub-families according to the type of ECDs they contain [21]. The most common ECD found in RLKs and RLPs is the leucine-rich repeat (LRR), but another seven ECDs have been reported, including lysin motifs (LysM), lectin, wall-associated kinases (WAK), S-locus domain, malectin-like, proline-rich, and cysteine-rich repeat [20]. As an example of gene amplification of RLKs and RLPs in Tracheophytes, 226 LRR-RLKs and 59 LRR-RLPs have been identified in the *Arabidopsis thaliana* genome, whereas only 81 and 8, respectively, have been found in the primitive vascular plant *Selaginella moellendorfii* [24]. On the other hand, in sharp contrast with the LRR subfamily, the LysM subfamily in *Arabidopsis* contains only 5 LysM-RLKs and 3 LysM-RLPs [13,24]. Noteworthy, the expansion in number and diversity of RLKs and RLPs reflects their co-evolution with their cognate MAMPs during land plant conquest [16]. It is important to emphasize that information regarding receptor types, specificities and responses are covered elsewhere [24]; thus, we will describe below the general picture of perception, activation, and early signaling mechanisms of RLKs and RLPs.

Upon MAMP recognition by RLKs and RLPs, these associate dynamically with co-receptors and receptor-like cytoplasmic kinases (RLCKs) to form a heteromeric complex (Figure 2B). These interactions occur in a ligand-dependent manner and, in particular, the specificity of RLCKs and their downstream targets is influenced by the configuration of the RLK complex [5,12]. Demonstrated co-receptors of RLKs are the LRR-RLKs BRASSINOSTEROID INSENSITIVE 1 (BRI1)-ASSOCIATED KINASE 1 (BAK1)/SOMATIC EMBRYOGENESIS RECEPTOR KINASE 3 (SERK3) (BAK1/SERK3) and BAK1-LIKE1/SERK4 (BKK1/SERK4) (Figure 2C) [26]. These co-receptors operate for several RLKs involved not only in regulating immunity but also in plant growth and development [26,27]. In the case of SERKs involved in plant immunity, these are not restricted to defense response against bacteria but also function against oomycetes and fungi [12,24]. On the other hand, LRR-RLPs and LysM-RLPs that do not carry a cytoplasmic kinase domain associate with RLKs to transmit the signal to downstream components [15]. Accordingly, LRR-RLPs and LysM-RLPs associate with the LRR-RLK SUPPRESSOR OF BIR 1-1/EVERSHED (SOBIR1/EVR) and the LysM-RLK CHITIN ELICITOR RECEPTOR 1/LysM-containing receptor-like kinase 1 (CERK1/LYK1) [28,29,30], respectively (Figure 2C). All these co-receptors are highly conserved in land plants and are crucial for PRR-mediated PTI [28].

In *Arabidopsis*, after perception of the bacterial flagellin peptide flg22 by the LRR-RLK FLAGELLIN SENSING 2 (FLS2) and BAK1, the RCLK BOTRYTIS INDUCED KINASE 1 (BIK1) forms a complex with FLS2 and BAK1 [28,31]. Given by auto- and trans-phosphorylation events of the complex, the association of receptor, co-receptor and RLCK amplifies the signal perceived in the apoplast by the ligand-receptor complex to the cell interior [32]. For instance, phosphorylation of BIK1, and related RLCKs such as BR-SIGNALING KINASE 1 (BSK1), BSK3, BSK5, avrPphB SENSITIVE 1 (PBS1), PBS1-LIKE 1 (PBL1), and PBL2, results in trans-phosphorylation of the phosphatase BRI1-SUPPRESSOR 1 (BSU1), triggering MAPK activation (Figure 2C) [33,34,35]. Two parallel MAPK cascades are activated upon MAMP perception, the first one being the module MEKK3/MEKK5-MKK4/MKK5-MPK3/MAPK6 and the second one consisting of the module MEKK1-MKK1/MKK2-MAPK4 [36,37,38]. MAPK cascades transduce the signal into the nucleus, resulting in transcriptional reprogramming [39,40,41]. Besides the activation of MAPKs, ROS production, cytosolic Ca^2+^ influx, and activation of CDPKs have also been found as outputs of PRR-mediated PTI (Figure 2C) [12,42]. ROS burst, for example, is achieved through the activation of membrane-resident RESPIRATORY BURST OXIDASE HOMOLOGs (RBOHs) by BIK1 (and related RLCKs) (Figure 2C) [43]. The Ca^2+^ influx, on the other hand, occurs through the activation of cyclic nucleotide-gated channel 2 (CNGC2) and CNGC4, which form a heteromeric calcium-permeable channel in response to phosphorylation by BIK1 and PBL1 (Figure 2C) [44,45]. BIK1 also phosphorylates and activates ion channels of the reduced hyperosmolality-induced [Ca^2+^] increase 1.3 (OSCA1.3) and glutamate receptor-like channels (GLRs) upon pathogen perception [46]. Oscillation of Ca^2+^ ions is detected by sensors containing EF-hand motifs that activate CDPKs (CDPK4, CDPK5, CDPK6, and CDPK11), which in turn decode the signal for mounting the proper plant response [47]. In the case of CDPK5, it has been involved in ROS activation after flg22 perception via the phosphorylation of RbohD [39,47]. ROS produced by RbohD induce Ca^2+^ release in neighboring cells, which activates CDPKs, allowing propagation of the signal from cell to cell (Figure 2C) [48]. Indeed, Marcec and Tanaka found recently that calcium has a significant impact on the initiation and amplification of the ROS signaling, whereas ROS is required for amplifying the calcium signal but not for its initiation [49].

Interestingly, RLKs and RLPs induce overlapping responses due to the activation of similar early and downstream signaling components [28]. One of these early signaling components is BAK1, a co-receptor recruited by LRR-RLPs RLP1, RLP23, RLP30, RLP32, and RLP42. In *Arabidopsis*, perception of NECROSIS- AND ETHYLENE-INDUCING PEPTIDE 1 (NEP1)-LIKE PROTEINs (NLPs) by RLP23 leads to its association with SOBIR1 and BAK1 and then signaling primarily via PBL31, which in turn confers resistance to bacteria, fungi and oomycetes [50,51]. The current model is that LRR-RLPs form a complex with SOBIR1 before ligand binding, and only upon ligand binding BAK1 is recruited to form an active receptor complex (Figure 2C) [28].

Despite the similar architecture of LRR-RLKs and LRR-RLPs, as well as the fact that BAK1 (and related SERKs) and BIK1 are recruited by both PRR types, Wan and co-workers found that RLP23- and RLP42-SOBIR1-BAK1 complexes lead to similar but distinct immune responses in timing and amplitude compared to LRR-RLK-mediated PTI [52]. Such differences are thought to be related to the phosphorylation status of BIK1, acting as a negative regulator in LRR-RLP-mediated PTI, which is in sharp contrast with its positive role in LRR-RLK-mediated PTI [52,53]. Furthermore, LysM-RLPs that bind chitin or bacterial peptidoglycan (PGN), i.e., LYM1, LYM3, and LYK5, form a complex with CERK1. Then, CERK1 phosphorylates the RLCK PBL27, which triggers the MEKK5-MKK4/MKK5-MAPK3/MAPK6 cascade (Figure 2C) [54,55]. Intriguingly, BAK1 is also involved in the LysM-RLP-mediated PTI since it phosphorylates CERK1 and increases its stability (Figure 2C) [56]. In parallel, BIK1 also acts downstream of CERK1 to regulate the chitin-induced ROS burst and callose deposition, similar to the mechanism described for LRR-RLKs (Figure 2C) [33]. In summary, recognition of MAMPs by PRRs, namely RLKs and RLPs, induces a common set of signaling and intracellular defense responses that lead to plant resistance against pathogens.

Finally, PRR complexes and their downstream signaling components must be under negative regulation to prevent spurious signaling. This regulation is achieved mainly through protein–protein interactions and post-translational modifications of key components in the PTI-mediated immunity. Among these negative regulators, BAK1-interacting RLK1 (BIR1), BIR2, and BIR3 act at the level of preventing the formation of active signaling complexes in normal conditions by sequestering BAK1 [57,58]. Accordingly, *bir* mutants display extensive cell death and activation of constitutive defense responses [59]. Another central signaling component in the PRR-signaling pathway is BIK1. Thus, BIK1 is negatively regulated, among other mechanisms, by proteasome-degradation mediated through the EXTRA-LARGE G PROTEIN 2 (XLG2) [60]. On the other hand, the plasma-membrane-associated host multifunctional protein RESISTANCE TO PSEUDOMONAS SYRINGAE 1 (RPM1)-INTERACTING PROTEIN 4 (RIN4) was also identified as a negative regulator of PTI because a mutation of this protein constitutively activates immune responses [61,62]. Activation of RIN4 occurs through phosphorylation by the RLCK RPM1-induced protein kinase (RIPK) (and likely by other RLCKs) upon MAMP recognition [63,64]. Another negative regulator is the Calcineurin B-like protein (CBL)-interacting protein kinase 6 (CIPK6), which inhibits the oxidative burst, as well as the MAPK cascade [65]. Transcription factors, such as CALMODULIN-BINDING TRANSCRIPTION ACTIVATORS 1/2/3 (CAMTA1/2/3), exert a negative control on the PTI-mediated transcriptional reprogramming [66,67,68]. Taken together, positive and negative regulation of plant innate immunity must be a well-balanced process, fine-tuning the amplitude, timing, and duration of immune responses to assure plant survival since immune responses temporarily suppress plant growth and vice versa.

## 3. Deconvoluting the NLR-Mediated Immunity: Perception, Activation, and Early Signaling Mechanisms

As described above, some components fulfill essential functions in the PTI pathway and, thus, it is expected to be targeted by microbial effectors. In fact, BAK1 and BIK1 are targets of unrelated effector proteins from bacteria, fungi, and oomycetes [69]. Additionally, pathogens deploy effectors to induce RIPK-mediated phosphorylation of RIN4 to suppress PTI. To protect these key components in the PTI-mediated immunity, plants have evolved intracellular NLRs that recognize pathogen effectors and trigger ETI [9,70]. Generally, this second type of plant immunity induces more robust and more prolonged immune responses than PTI, conferring complete and specific resistance to viruses, bacteria, fungi, nematodes, and insects [10].

NLR-coding genes are one of the largest and most variable gene families found in plants [71,72]. Ancient NLRs first appeared in green algae and then diversified through gene duplication and domain fusions in early land plant lineages [73,74]. A typical NLR consists of a central nucleotide-binding site (NBS) with characteristic motifs widely conserved in plants, whereas its N- and C-terminal regions are highly variable [75]. According to the N-terminal domains, NLRs are classified into three subfamilies: Toll/Interleukin-1 receptor/Resistance (TIR)-NLRs (TNLs), coiled-coil (CC)-NLRs (CNLs), and Resistance to Powdery Mildew 8 (RPW8)-like CC domain-NLRs (RNLs) (Figure 3A) [71,76]. Whereas TNLs are restricted to dicots, CNLs are found in dicots and monocots [71,77]. Importantly, RNLs are relatively conserved across land plant species, namely Bryophytes and Tracheophytes [22,78]. On the other hand, the C-terminal LRR domain is composed of a tandem repeat of amino acids and is variable in length [75,79]. Similar to the case of PRRs, it is the LRR domain in NLRs that supervises the presence of pathogens through direct interactions with microbial effectors or, indirectly, via monitoring molecules modified by effectors [80]. Whereas the LRR domain of NLRs functions in the recognition of effectors, the NBS binds ADP in the resting state and ATP in the active state, acting as molecular switches that control signal transduction. In short, ADP/ATP exchange at the NB domain produces conformational changes that lead to the formation of NLR oligomers. Such oligomerization triggers Ca^2+^ influx, ROS burst, hormone signaling, transcriptional reprogramming, and programmed cell death, also known as hypersensitive response (HR) [9,81].

Two major types of plant NLRs have been identified: those that function as direct or indirect sensors of pathogen effectors (TNL and CNL members) and those that act as NLR helpers (RNL members) (Figure 3B). Recognition of effectors by plant NLRs follows multiple mechanistic models (direct, guard, decoy, and integrated decoy) described elsewhere [24,25,82] and is beyond the scope of this review. Briefly, when NLRs work as singletons, these recognize microbial effectors directly or indirectly and trigger downstream immune responses, behaving both as a sensor and as a signal inducer (Figure 3B). On the other hand, when NLRs function as a pair or network, one of them perceives the effector (NLR sensor), whereas the other plays a signaling role (NLR helper) (Figure 3B) [76,77]. Given that pathogens are more likely to acquire new mutations by their higher population sizes and short generation times, indirect NLR-mediated effector recognition has been selected during plant evolution to counteract the evolutionary imbalance with their pathogens. Hence, indirect recognition of effectors imposes on pathogens a higher fitness cost than the scenario of simply avoiding NLR-binding when the effector recognition is direct [83]. In that sense, according to indirect mechanistic models, a broad pathogen-recognition spectrum is achieved through a limited repertoire of plant NLRs by sensing effector-mediated modifications of a host target protein, namely the guardee or the decoy. It has been hypothesized that NLRs may have evolved from multifunctional singleton proteins to functionally specialized and interconnected receptor pairs and networks [84].

For example, the *Arabidopsis* CNL HOPZ-ACTIVATED RESISTANCE1 (ZAR1) is a singleton NLR that recognizes multiple pathogen effectors [86]. ZAR1, together with the RLCK Resistance-related KinaSe 1 (RKS1), detect multiple effectors by monitoring the RLCK PBL2. The interaction of RKS1 with ZAR1 maintains the NLR in a monomeric inactive form. Modification of PBL2 by effectors leads to its association with the ZAR1-RKS1 complex, inducing a conformational change in RKS1 that leads to the activation of the ZAR1-RKS1-PBL2 complex and its subsequent oligomerization to form an ion channel [87,88,89]. Overall, the ZAR1-mediated ETI results in cytosolic calcium influx, ROS accumulation, kinase cascade and HR (Figure 3C) [87]. Whether the pore-forming function of other CNLs and TNLs is conserved, remains to be explored. Other singleton CNLs such as RPM1 and the RESISTANT TO PSEUDOMONAS SYRINGAE 2 (RPS2), are essential for ETI-mediated immunity since they monitor RIN4, a key component of the PTI-mediated immunity that is targeted by various effectors [90,91,92]. When microbial effectors modify RIN4, this leads to the activation of RPM1 or RPS2, respectively [63,93,94]. Currently, it is unknown how the different modifications on RIN4 activate RPM1 or RPS2, but the outcome, namely the oligomerization to form a functional resistosome, may be similar to ZAR1. Noteworthy, many steps downstream of the recognition of effectors by CNLs remain a mystery, but some key players have been found. For instance, most CNLs require a protein localized in the plasma membrane for ETI, the so-called NON-RACE-SPECIFIC DISEASE RESISTANCE 1 (NDR1) [95]. However, it is unclear if NDR1 is required for the pore-forming activity of ZAR1 (or other CNLs). Since NDR1 shares similarities to mammalian integrins [96], it is likely that, upon pathogen attack, NRD1 and CNLs work together to signal any change in the plasma membrane or the extracellular matrix. Future studies will clarify this matter.

Although some singletons appear to act alone, other NLRs work in genetically linked pairs, a fast-evolving sensor NLR (TNL and CNL members) and a slow-evolving NLR helper (RNL members), suggesting that opposite selective forces have driven the evolution of paired NLRs (Figure 3B) [97]. RNL helpers are subdivided into two subclades, namely ADRs and NRGs, which are given by the homology of their RPW8 domain to the ACTIVATED DISEASE RESISTANCE 1 (ADR1) of *A. thaliana* or the N REQUIREMENT GENE 1 (NRG1) of *Nicotiana benthamiana*, respectively [76]. Both RLN helpers are required by sensor NLRs for immune activation, primarily by TNLs [77,98,99]. Like CNLs, TNLs also oligomerize following effector recognition (Figure 3D) [100,101], but in this case, their oligomerization activates a NADase activity, producing adenosine diphosphate ribose (ADPR) and a variant of cyclic ADPR (v-cADPR). Such small molecules are required for downstream signaling via the Enhanced Disease Susceptibility 1 (EDS1), a pivotal regulator of ETI that transduces signals to turn on transcriptional defense response and HR (Figure 3D) [85,102,103,104]. Besides EDS1, Phytoalexin-Deficient 4 (PAD4) or Senescence-Associated Gene 101 (SAG101) are also important regulators of ETI and basal immunity [85,105,106]. These proteins are characterized by possessing an N-terminal lipase-like domain (LLD) and a unique C-terminal known as EP-domain (EPD), thereby they are also known as EPD containing proteins (EP-proteins) [85,105,107,108]. EDS1 forms exclusive heterodimers with PAD4 or SAG101, which in the context of NLR-mediated immunity, two modules have been found: the EDS1-PAD4-ADR1 and EDS1-SAG101-NRG1, which are not functionally interchangeable (Figure 3D). Via TNLs and some CNLs, EDS1-PAD4-ADR1 promotes mainly pathogen restriction and contributes weakly to HR. On the other hand, EDS1-SAG101-NRG1, which signals downstream of TNLs, promotes HR and contributes weakly to pathogen restriction (Figure 3D) [106,109,110]. Although the mode of action of these complexes remains unknown, the recent finding that ADR1 and NRG1 function as calcium channels [111] allows us to speculate that the association and activation of EDS1-PAD4-ADR1 or EDS1-SAG101-NRG1 result in a calcium influx that triggers downstream immune responses. It is also conceivable that the cavity formed between the heterodimers (EDS1-PAD4 or EDS1-SAG101) could serve as a binding pocket for unknown proteins or ligands that activate immune signaling [102,112]. Some TNLs rely on their downstream signaling through both signaling modules. For example, the TNL pair composed of the RESISTANCE TO RALSTONIA SOLANACEARUM 1 (RRS1)/ RESISTANCE TO *PSEUDOMONAS SYRINGAE* 4 (RPS4) functions with the ADR1 family to achieve a transcriptional reprogramming of defense genes and pathogen resistance but also works with NRG1 to trigger HR [77,99,113,114]. It has been speculated that TNLs transduce their signal via ADR1s, as well as NRG1s, likely to provide resilience against interference by effectors [115]. In addition, the finding that ADR1 mediates the biosynthesis of salicylic acid (SA) biosynthesis downstream of RPS2, but independent of its oligomerization capacity, reveals that ADR1 works beyond its function as a calcium channel [82].

In summary, CNLs and TNLs seem to signal through NDR1 and EDS1, respectively, which in the case of EDS1, branches further in two distinct downstream signaling modules formed by EDS1-PAD4-ADR1 and EDS1-SAG101-NRG1. Although early mechanisms of CNL- and TNL-mediated ETI appear to be different, both assemble into higher-order resistosomes after effector recognition and share the property of increasing Ca^2+^ levels in the cytoplasm [111,116]. The pivotal role of Ca^2+^ is supported by the finding that Ca^2+^ channel blockers inhibit cell-death-inducing activities of CNLs, TNLs, and RNLs [117]. Finally, an important matter that is emerging is how plants keep NLRs in check. This is summarized by Lapin and co-workers, in which transcriptional, posttranscriptional, and posttranslational mechanisms are involved to constrain ETI [115].

## 4. PTI and ETI as One Integrated Pathway

Even though the molecular mechanisms of the NLR-mediated immunity are not fully understood yet, ETI triggers similar immune responses as PTI. Such common responses include calcium influx, ROS generation, kinase cascades, hormone production, callose deposition, and transcriptional reprogramming [10]. In this regard, recent studies have reinforced the picture that these two pathways are interdependent and converge at multiple nodes, potentiating each other to defend successfully against microbial infections. Below we highlight those shared components and signal outputs that intimately link both plant immunity branches.

### 4.1. Inherent Relations between PTI and ETI: Plant Immunity and Cell Death

Identification and characterization of several lesion mimic mutants (LMM) have been a powerful tool in the quest to unravel how plant immunity works. Hallmarks of LMMs comprise an accumulation of ROS, activation of SA-mediated defense responses, and spontaneous HR-like cell death, all of them reminiscent of the NLR-mediated immunity [81,118]. Intriguingly, deletion of PRR-signaling components, such as BAK1, BKK1, BIK1, CNGC2/4, RBOHD/F, MEKK1, MKK1/2, MAPK4, and CAMTA3, exhibit autoimmune phenotypes characterized by HR-like cell death [59,119,120]. The autoimmunity observed in some of these mutants is caused by the activation of multiple NLRs [121,122,123]. Accordingly, the activation of NLR sensors is a consequence of modifications or deletions of the monitored cellular component (guardee or a decoy), which is either a key node within the PTI- or the ETI-mediated immunity [124,125]. For instance, the HR-like cell death observed in *bak1 bkk1* is caused by the activation of NLR-mediated ETI rather than the reduction of PTI [126,127]. Moreover, the finding that the LMM phenotype of *bak1 bkk1* is dependent on the ADR1 suggests that BAK1 and BKK1 are guarded by a yet unidentified NLR sensor [12,128]. This is consistent with the compromised ETI-associated pathogen restriction previously found in *bak1 bkk1* against *Hyaloperonospora arabidopsidis* [129]. Downstream of BAK1/BKK1 is found BIK1, which phosphorylates the CNGC2/CNGC4 channel and RbohD, leading to elevated cytosolic Ca^2+^ and ROS burst upon MAMP recognition, respectively [44]. Interestingly, RBOHD was recently found to be necessary for the response against *Pseudomonas syringae* DC3000, suggesting that this PRR-triggered ROS gene acts as a central hub that links PTI and ETI [130]. Yuan and co-workers further demonstrated that phosphorylation of RBOHD in Ser343 and Ser347 by BIK1 during ETI occurs in a BAK1- and BKK1-dependent manner [31,131,132,133]. This suggests that BIK1 is another point of integration between PTI and ETI.

On the other hand, incomplete loss of the PRR-derived Ca^2+^ influx and ROS burst in the *cngc2 cngc4* mutant suggested the involvement of additional Ca^2+^ channels in PTI signaling [134]. Interestingly, the LMM phenotype of *bak1 bkk1* is a consequence of the misregulation of two additional calcium channels, CNGC19 and CNGC20 [135]. Whereas phosphorylation by BIK1 stabilizes CNGC19/CNGC20, phosphorylation of CNGC20 by BAK1 negatively regulates its abundance; thus, BAK1-mediated homeostasis of CNGC19/CNGC20 is key for precise control of plant cell death [135]. Indeed, the recent finding that a recessive gain-of-function CNGC20 mutant (cngc20-4) partially restores disease resistance in *eds1* suggests that CNGC20 is a convergent node between PRR- and NLR-mediated immune pathways [135]. Since high levels of cytosolic Ca^2+^ are responsible for the autoimmunity observed in *cngc20-4* [135], it is likely that cytosolic Ca^2+^ might enhance the activation of CDPKs such as CDPK5, which directly phosphorylates and regulates RBOHD for ROS burst [47]. In addition, cytosolic Ca^2+^ might also activate the master transcription factor CALMODULIN-BINDING PROTEIN 60 g (CBP60g), which together with the SYSTEMIC-ACQUIRED RESISTANCE DEFICIENT 1 (SARD1), upregulate genes involved in SA synthesis and signaling, such as Isochorismate Synthase 1 (ICS1), EDS5, AVRPPHB SUSCEPTIBLE 3 (PBS3) and NON-EXPRESSOR OF PATHOGENESIS-RELATED GENES 1 (NPR1), as well as genes encoding key components of the ETI-mediated immunity, such as EDS1 and PAD4 [136,137,138,139,140]. Such a scenario is supported by the discovery that simultaneous removal of ICS1 and EDS1 completely suppresses cngc20-4 autoimmunity [135]. Further, the fact that PTI-induced SA accumulation is almost completely depleted in *cbp60g sard1* double-mutant implies that both immune-regulating transcription factors are key for hormone production in both PTI and ETI [139,141]. Altogether, CNGC19/CNGC20 seems to play a putative central role in PTI and ETI, thereby a guard protein should protect this hub, but this remains to be determined. Moreover, the involvement of CNGC2/CNGC4 in RPS2-mediated cell death pinpoints to a synergistic interaction between NLRs and membrane-localized calcium channels [134,135,142]. More work is needed to solve the puzzle and uncover precisely how PRR-signaling components control Ca^2+^ levels and activate ETI responses.

### 4.2. EDS1-PAD4, a Convergence Hub between PRR- and NLR-Mediated Immune Responses

HR is a hallmark response of ETI. The finding that TNL-mediated HR is compromised in PRR/co-receptor mutants suggests that TNLs rely on PRR signaling components [133]. This dependency of TNLs on PRR signaling components is further confirmed by the discovery that activation of multiple NLRs promotes transcript and protein accumulation of BAK1, SOBIR1, BIK1, RBOHD, and MAPK3 [130]. However, PTI also appears to be dependent on ETI-signaling components; thus, the influence between PTI and ETI seems to be mutual [132,143,144]. Compelling evidence published by Pruitt et al. and Tian et al. showed that EDS1, PAD4, and ADR1 are required for PTI activation, particularly in RLP-initiated signaling (Figure 4) [145,146]. Briefly, *Arabidopsis* plants defective in the EDS1-PAD4-ADR1 module showed diminished RLP23-dependent induction of ethylene production, ROS burst, callose deposition and resistance against *Pseudomonas syringae* pv. Tomato (Pst) [146]. The impaired RLP-signaling is not because of regulatory effects of EDS1-PAD4 on PRR signaling components since their transcriptional or translational levels were unaffected by the lack of EDS1 and PAD4 [146]. In addition, pools of SOBIR1 and PBL31 were found in close proximity with EDS1-PAD4 and ADR1 in the plasma membrane, suggesting that EDS1-PAD4-ADR1 could function immediately downstream of RLP23-SOBIR1 [146]. Consistently, Tian and co-workers found that RLK- and RLP-mediated PTI is particularly dependent on EDS1, PAD4, and ADR1 (Figure 4) [145]. For instance, defense genes expression, SA biosynthesis and resistance to virulent pathogens were greatly diminished in the triple mutant eds1-24 pad4-1 adr1 and modestly reduced in *eds1-24 sag101-1 nrg1* plants [145]. Furthermore, activation of PTI can upregulate the expression of a number of TNLs; however, such upregulation is dependent on Ca^2+^ influx but not on ROS [145]. Finally, the overexpression of some of these PTI-induced TNLs resulted in increased defense outputs in *Arabidopsis* and *N. benthamiana*, suggesting that PTI triggers immune responses through TNL signaling pathways [99,147,148,149]. For a time, it was known that the EDS1-PAD4 complex was involved in basal immunity response, but the molecular mechanism was unclear [105,150,151]. Now that RLK- and RLP-mediated PTI have been found to be dependent on EDS1-PAD4-ADR1 (Figure 4), it is conceivable that basal immunity might be achieved through surface-localized PRRs. Taken together, these findings indicate that EDS1-PAD4-ADR1 is an important node linking PTI and ETI in *Arabidopsis* and, likely, in all dicot plants.

#### RNL Helpers Co-Evolved with EDS1/PAD4/SAG101

The helper NLRs, ADR1, and NRG1, represent an ancient branch of CNLs that occurs in widely diverged plant lineages [78,152], whereas members of the EDS1 family (EDS1/PAD4/SAG101) are found only in seed plants, but not in non-seed plants or algae [108,114,153,154]. Consistent with a helper function rather than a sensor role, RNLs usually display low copy number and high conservation in plant genomes [78]. Importantly, phylogenomic studies have shown that the occurrence of EDS1 and PAD4 correlates with the distribution of CNLs and ADR1s, whereas plants lacking SAG101 and NRG1 also lack TNLs [22,105,114,155]. This is consistent with genetic, biochemical, and structural studies showing a co-evolved functional relationship between TNLs, EDS1/PAD4/SAG101, and RNLs, further reinforcing the current picture that, downstream of EDS1/PAD4/SAG101, specific RNLs play a role in TNL immunity [77,109,155,156].

Although it is generally assumed that members of the EDS1 family and RNLs are absent in monocots [74], a recent survey on early monocot lineages identified transcripts orthologous to EDS1, PAD4, RNL genes, and NDR1, suggesting that these components were present in the most recent common ancestor of monocots [154]. Interestingly, a comparative approach identified genes that were convergently lost with EDS1, PAD4, ADR1, NRG1, and NDR1 in plant species inhabiting water-saturated environments [154]. Such genes, known as AngioSperm Typically Retained, EDS1-Lost (ASTREL), were analyzed at the level of gene expression in *Arabidopsis* and *Oryza sativa*, finding that ASTREL genes were responsive not only to pathogen infection but also to drought stress [154]. This supports previous studies in which drought tolerance was found to be dependent on EDS1 and PAD4 [157,158,159]. On the other hand, despite the absence of TNLs, most monocots retain EDS1, suggesting that it might be required for either CNL signaling or pathogen-triggered expression of conserved TIR-only genes [115,145,148,160,161]. Conversely, TNL genes exist in seed and non-seed plant species without SAG101 and NRG1 [74,162], indicating that some TNLs signal without SAG101 and NRG1. Altogether, these findings suggest that plants have rewired components of plant immunity to defend successfully against microbial infections and, more importantly, that some components of plant immunity have formed a functional alliance that goes beyond biotic stress response. Finally, it is tempting to speculate that the involvement of EDS1-PAD4-ADR1 in linking PTI and ETI in *Arabidopsis* could be extrapolated to other plants in which this module is conserved, namely angiosperms and gymnosperms. It is likely that EDS1-PAD4-ADR1 represents an ancient module within seed plants that promotes pathogen restriction through surface-localized PRRs, whereas EDS1-SAG101-NRG1 is a recent acquisition specifically involved in HR (See Figure 3D).

### 4.3. ROS and SA Are Common Outputs of PTI and ETI

Whereas mechanisms of activation of PRR- and NLR-proteins are relatively known, the picture of downstream signaling mechanisms leading to the restriction of the pathogen is less understood. In ETI, HR-conferred resistance is preceded by a series of biochemical and cellular signals, then followed by systemic acquired resistance (SAR) of the host [163]. The role of ROS and SA in HR-conferred resistance depends on precise spatiotemporal regulation of these molecules for an effective pathogen arrest.

PTI and ETI share several signals, including ROS and SA. In both cases, an increase in SA levels is preceded by apoplastic ROS bursts mediated by NADPH oxidases (NOXs), such as RBOHD [164,165,166]. Given that increase of cytosolic Ca^+2^ is upstream of apoplastic ROS production during ETI [167], together with the notion that NLR-triggered HR is calcium-dependent, it is conceivable that a Ca^+2^ signal mediates activation of SA production triggered by ROS [130,131,132]. Emerging studies also have revealed that SA plays a role in modulating ROS homeostasis; thus, SA and ROS have intertwined roles in plant response to biotic stress.

#### 4.3.1. RBOHD-Dependent ROS Production

Besides its role as direct weapons against pathogens, ROS are signaling molecules as well, triggering HR at the site of infection to limit pathogen progression [164,168,169,170]. Hydrogen peroxide, for instance, promotes callose deposition and restricts fungal and oomycete infection through the activity of peroxidases that cause protein and phenolic cross-linking to reinforce the cell wall [171,172]. Whereas PTI induces a fast and transient ROS burst, ETI is associated with a biphasic ROS burst [130,132,173,174]. The first ROS burst occurs within minutes after infection, and is transient and with low amplitude, whereas the second one is initiated a few hours after infection and it is sustained and with high amplitude [130,132]. Moreover, the first one is mostly apoplastic, tightly linked to posttranslational activation of RBOH, whereas the second ROS burst is produced by different cellular compartments (apoplasts, chloroplasts, mitochondria, and peroxisomes), requiring the activation of RBOH at the transcriptional level [174,175]. Since the RBOHD-dependent ROS production uncovered by Yuan et al. and Ngou et al. requires PRR-signaling components for the second burst during ETI [130,132], their studies demonstrate a tight synergistic interaction and integration of signals from both PTI and ETI for a successful response against microbial pathogens. Whether PRR-signaling components such as BAK1, BKK1, or BIK1 are direct targets of NLRs for RBOH activation or how chloroplast-derived ROS is coordinated with PRR- and NLR-signaling remains to be uncovered in the near future.

#### 4.3.2. SA at the Central Core of Metabolic Defense Strategies

In addition to ROS, SA has been identified as one of the key components of immune signaling [176,177]. Derived from the shikimate-phenylpropanoid pathway, SA quickly accumulates at the infection site, triggering a plethora of immune responses such as massive transcriptional reprogramming, cell wall strengthening, and production of secondary metabolites and antimicrobial proteins [140]. SA is required for an effective defense response against biotrophic and hemibiotrophic pathogens, not only in ETI but also in PTI and SAR [178,179,180]. SAR is defined as the induction of immune responses in uninfected parts of the plant, characterized by the activation of Pathogenesis-Related (PR) genes and proteins with antimicrobial activity [181,182,183].

In *Arabidopsis*, most of the pathogen-induced SA is produced through ICS1, EDS5, and PBS3, known as the ICS pathway [140,180]. ICS1 converts chorismate to isochorismate in plastids, which is then transported to the cytosol by EDS5. Finally, PBS3 and ENHANCED PSEUDOMONAS SUSCEPTIBILITY 1 (EPS1) catalyze the final conversions to SA [184,185]. Importantly, both NDR1- and EDS1/PAD4-signaling are involved in SA accumulation [186,187,188]. Thus, SA-mediated immune responses are mostly mediated by TNLs, and also by some CNLs, during ETI. In that sense, transcriptional regulation of ICS1, as well as SA metabolic genes, is dependent on EDS1/PAD4 [108,189,190]. Recent findings suggest the involvement of CBP60g and SARD1 in the activation of ICS1, EDS5 and PBS3 [135], implying that cytosolic Ca^2+^ burst is upstream of SA.

On the other hand, responses downstream of SA are mediated by NPR1, a pivotal regulator of SA signaling that functions as a cofactor of TGACG (TGA) transcription factors to activate PR genes, including PR1, PR2, and PR3 [191,192,193]. SA promotes the activity of NPR1, as well as its transcriptional activation, to induce the expression of defense-related genes [194]. Intriguingly, CBP60g and SARD1 also positively regulate NPR1, EDS1 and PAD4 [136,137,138,139,141], reflecting an intricate relationship between these immune regulators during SA production. Moreover, SA enhances the expression of EDS1 and PAD4, indicating a feedback loop to further amplify the plant defense response [186,195,196]. How PRR-/NLR-signaling components, membrane-localized calcium channels (i.e., CNGC2/4 and CNGC19/20) and CDPKs coordinate to trigger SA-mediated immune responses is something that still remains elusive and needs to be addressed. Finally, the observation that the SA pathway is conserved in species that have lost EDS1, PAD4, SAG101, NDR1, and RNL genes, suggests that the role of SA is ancient, operating in the defense system of non-vascular land plants [154,176,197,198].

#### 4.3.3. Alternative Routes Provide Resilience of Plant Immunity

SA biosynthesis and signaling pathways are targeted by pathogen effectors [199]. Therefore, an important evolutionary mechanism for avoiding pathogen interference is the property of network robustness in plant immune signaling [178]. Accordingly, plants synthesize N-hydroxy-pipecolic acid (NHP), an immune-activating metabolite required for local and systemic immunity [200,201,202]. AGD2-LIKE DEFENSE RESPONSE PROTEIN 1 (ALD1), SARD4, and FLAVIN-DEPENDENT MONOOXYGENASE 1 (FMO1) are pathogen-inducible genes that also depend on EDS1/PAD4-signaling for NHP biosynthesis [201,203,204]. Similar to the case of the SA biosynthetic genes, the NHP biosynthetic genes ALD1, SARD4, and FMO1 are regulated via CBP60g and SARD1 [139,141,180,204]. Importantly, SA and NHP synergistically influence each other. SA can induce NHP accumulation and vice versa [201,205,206,207]. Both SA induction of NHP biosynthetic genes and NHP induction of SA-associated genes depend on NPR1 [194,203,208], showing that SAR is dependent on mutual amplification of these immune-activating metabolites.

Regarding MAPK activation, it is reminiscent of the biphasic behavior of ROS during PTI and ETI [132], namely that PTI triggers a fast and transient activation of MAPKs, whereas ETI results in sustained MAPK activation [209]. Transient MAPK activation, involving MAPK3/6 during PTI, results in the induction of SA-dependent SA-responsive genes. On the other hand, sustained activation of the MAPKs during ETI supports the expression of SA-responsive genes in a SA-independent manner [209]. These data, together with the knowledge that the biosynthetic precursor of NHP, the pipecolic acid (Pip), can activate MAPKs in a BAK1- and BKK1-dependent manner and also that the Pip biosynthetic genes are required for the sustained MAPK activation during ETI [130,206], pinpoint to the role of MAPK cascades as a convergent point of PRRs and NLRs to trigger SA- and NHP-mediated defense responses. Finally, the identification of an EDS1/PAD4 branch that functions independently of ICS1-generated SA suggests that plants have evolved alternative routes for preserving SA-induced defense responses against pathogen or genetic perturbations [209,210,211]. Such compensatory relationships between SA, NHP, MAPKs, and EDS1/PAD4 confer robustness of the immune signaling network.

## 5. Plant Immunity in the Light of Evolution

Most of what we know about the plant immune system derives from studies on evolutionarily young flowering seed plants (angiosperms). In contrast, we know almost nothing about the perception, signaling, and defense-action layer in early-diverging land plants [3,212,213]. Therefore, a comprehensive understanding of plant immunity from an evolutionary viewpoint is urgently needed. Below, we offer first an overview of the conservation of canonical components connecting PTI and ETI by using plant species that represent evolutionary snapshots within the plant tree. Then, information about EvoMPMI studies that offer some insights regarding the ancestral state of plant–pathogen interactions is described in a broad-brush manner, attempting to highlight those commonalities in defense responses between early-diverging land plants and angiosperms.

### 5.1. A Snapshot of Canonical Components Shared between PTI and ETI, So Far

As described above, BAK1, BKK1, SOBIR1, BIK1, MAPKs (3/4/6), RBOHD, CNGCs, the EDS1 family, RNLs, CBP60g, SARD1, and NPR1 have been found to be important nodes in connecting both branches of plant immunity. To start understanding how these components were recruited in a stepwise manner, it is necessary to have an overview of their conservation along the plant kingdom. Thus, in an attempt to discover the evolutionary origins of these essential components in plant immunity, a comparative approach was carried out (Figure 5). To identify putative orthologs of these proteins, reciprocal BLAST searches using blastp were performed between *Arabidopsis* protein sequences and plant proteomes representing evolutionary snapshots in the plant tree [214,215,216], whereas the conservation of other components was taken from published works [1,4,154,217,218]. To identify the most likely ortholog for each of our selected canonical components, a domain-oriented orthology inference approach was utilized [219], meaning that besides identifying the best BLAST hit, its domain architecture was analyzed with the NCBI’s Conserved Domain Database [220,221]. In that sense, putative orthologs of BAK1, BKK1, SOBIR1, BIK1, MAPKs (3/4/6), RBOHD, CNGCs, the EDS1 family, RNLs, CBP60g, SARD1, and NPR1 were identified (Figure 5). In cases in which a component was not found due to a poorly annotated genome for a given plant, but there was a BLAST hit in an early-diverging plant, we follow the parsimony principle, inferring the existence of an ortholog in such a plant. Given the widespread conservation of PRRs and NLRs in Streptophytes, they were used as reference points (Figure 5).

Overall, the structural similarity of components connecting PTI and ETI and thus their conservation is highly correlated with proximity in the phylogenetic tree, whereas components of early-diverging plants showed less conservation (Appendix A). Our comparative approach reflected the known functional coevolution of EDS1/PAD4/SAG101 and RNLs (Figure 5). For instance, the lack of SAG101 correlated with the absence of NRG1 in monocots [105,108,109,154]. More importantly, by gathering and integrating the evolutionary conservation of signaling and executors connecting PTI and ETI allowed us to start inferring their putative origin and recruitment as part of the plant immunity network. Hence, it is clear that the EDS1/PAD4/SAG101 module is a recent innovation, likely exclusive for seed plants, whereas early signaling components such as BAK1 and BIK1, as well as their downstream targets for Ca^2+^ and ROS bursts, are more ancient (Figure 5). On the other hand, the network of SA production, perception and signaling has been relatively conserved since the emergence of land plants (Figure 5), suggesting that the function of SA in plant immunity is ancient. Altogether, the overall picture regarding the conservation of these canonical components suggests that, along plant history, they were recruited in a stepwise manner, likely by forming simple interaction units at the beginning and then, complex networks as found currently in modern plants. Future studies in early-diverging plants will illuminate which of these components constitute the core of the defense signaling network and which constitute lineage- and/or environment-specific circuits.

### 5.2. Ancient Host-Microbe Interactions

Bacteria and fungi emerged long before the common ancestor of plants [223,224]. Hence, plants have encountered and established a wide range of interactions with microorganisms since their emergence, from mutualism to parasitism. The appearance of visible disease symptoms is merely an extreme outcome of this continuum of plant–microbe associations.

Likely by the lack of mechanical protection typical of the vascular plants, the common ancestor of land plants and its subsequent early lineages were equipped with a biochemical capability as a part of their survival strategy [218,225]. For example, a major source of plant secondary metabolites is the phenylpropanoid (PP) pathway [226], from which derives phenylpropanoids as major players in plant immunity. Both SA and phenylpropanoids share some steps in their production process. This connection occurs in the second route of SA production, which also starts from chorismate but goes through intermediates, being one of them the amino acid phenylalanine, a substrate of phenylalanine ammonia-lyase (PAL), hence the name PAL pathway [184,185,227]. According to their ancient origins, PAL and ICS are conserved in some algae but mostly in all land plants, from bryophytes to pteridophytes to higher plants [218].

In *Marchantia polymorpha*, an early-divergent land plant lineage, RLKs, NLRs, and SA pathway genes have been found [222,228]. In fact, a conserved biochemical defense response has been found in *M. polymorpha*, namely the role of PP-associated metabolites in mitigating pathogen infection [229]. Moreover, using this plant pathosystem, Gimenez-Ibanez and co-workers show that the mechanisms governing plant–microbe interactions described in angiosperms are ancient and conserved in liverworts [197]. This is in line with previous studies showing that in *Physcomitrium patens*, another early-diverging land plant, the presence of pathogens is perceived, and a defense response is activated, characterized by increased production of ROS, induction of a hypersensitive-like response, cell wall reinforcement, and changes in hormones levels, as well as transcriptional reprogramming [212,230,231,232]. For example, the perception of fungal chitin in *P. patens* is carried out by an ortholog of CERK1, which relies on MAPK4 to transduce the signal for the activation of defense genes [233,234]. On the other hand, incipient studies in ferns show that the application of SA or pathogen challenge also activates PR genes [235], indicating that the SAR-like reaction is evolutionarily conserved between bryophytes, seedless plants, and angiosperms [236].

Although research on bryophytes is emerging, lycophytes and pteridophytes have yet to catch up. The recent field of EvoMPMI has emerged to patch gaps over the evolutionary timescale of plant history [237], but much work is still needed to uncover commonalities and differences of plant–pathogen interactions in nonvascular and vascular plants. Such efforts will bring light to the functional understanding of how immunity pathways operated in early-diverging land plants, but most importantly, what was the basic structure of the plant immune system when plants emerged and conquered the land around 500 MYA.

## 6. Concluding Remarks

Since their emergence, plants have been engaged in a constant battle against microbes. Over time, plants became equipped with sophisticated defense mechanisms, whereas pathogens developed an arsenal of counter strategies to overcome them. Accordingly, plants evolved two layers of plant immunity, an ancient form of innate immunity known as PTI, which constitutes the first layer, and ETI as the second one, which is highly specific and confers complete resistance to a wide range of pathogens. Traditionally, PTI and ETI have been viewed as independent pathways; however, given the recent findings, the distinction between them is blurring, viewed now as a continuum. Here, we have provided an overview of PTI- and ETI-mediated immunities, highlighting those converging nodes between them at their early signaling mechanisms, as well as their commonalities in the defense-action layer. The general picture is that PTI and ETI are interdependent and potentiate each other to defend successfully against microbial infections. Particularly, we offered an evolutionary perspective regarding those key components connecting PTI and ETI, emerging from it that the current network of plant immunity is the outcome of stepwise recruitments and losses of components. Future research will add more components shared by both layers of immunity. On the other hand, more EvoMPMI studies are needed to uncover the core immunity network of the ancient plant defense system. By exploring and harnessing the molecular mechanisms of defense responses employed by early-diverging plants may extend the possibilities for engineering and generating pathogen-resilient crops.

## Figures and Tables

**Figure 1 ijms-23-12974-f001:**
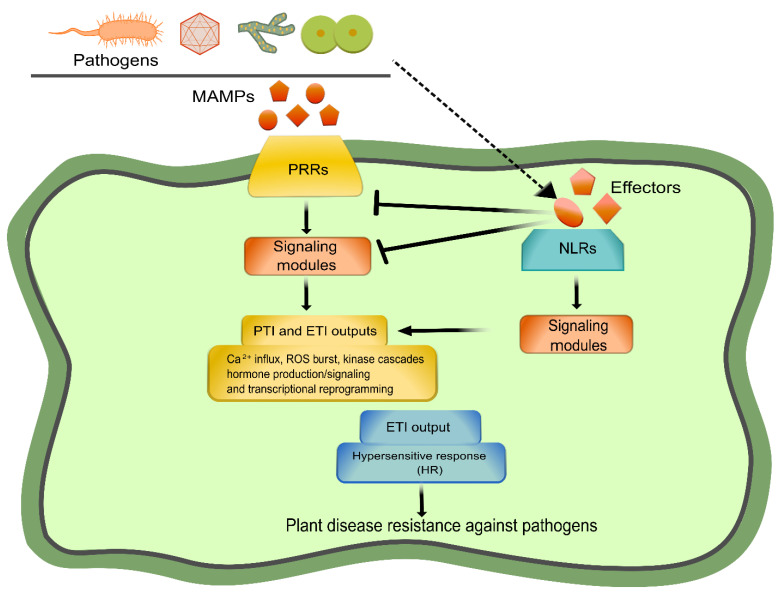
Plant immunity branches: PRR-mediated PTI and NLR-mediated ETI. A general model depicting the two layers of plant immunity is shown. The activation of both pathways can be divided into three phases: immune perception, signal integration, and defense execution. PRRs and NLRs are involved in the first phase, recognizing pathogen-derived molecules. Then, the signal is transduced and integrated through a network of components (signaling modules), deriving in downstream outputs (ROS burst, hormone production/signaling, transcriptional reprogramming, and cell death) aimed at restricting pathogen invasion. Created with Inkscape (https://inkscape.org) and GIMP v2.10.24 (https://www.gimp.org).

**Figure 2 ijms-23-12974-f002:**
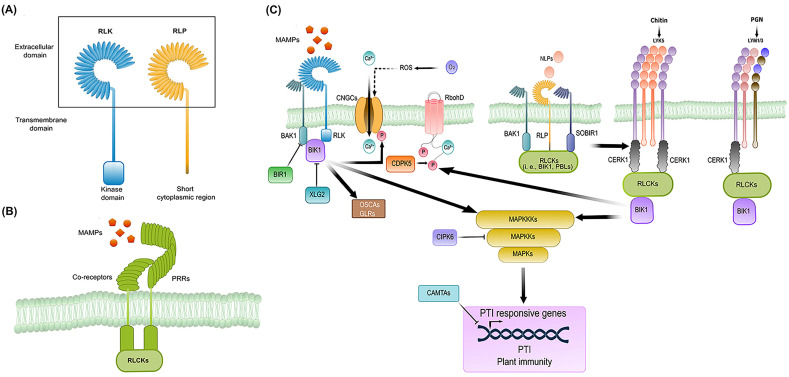
An overview of PRR-mediated PTI. (**A**) Classification of PRRs as RLKs and RLPs according to their structural differences in the intracellular region. (**B**) Ligand-dependent interactions of PRRs with co-receptors and RLCKs. (**C**) An outline of the PRR-triggered signaling cascade that leads to PTI is depicted. At the level of early signaling, BAK1, SOBIR1, and CERK1 are canonical co-receptors of PRRs (RLKs and RLPs). Then, the signal is transduced and integrated by RLCKs to downstream modules such as kinases, RBOHs, CNGCs, among others. See the main text for details. Section 2A and part of 2C were adapted with permission from [20,25], respectively. Created with Inkscape (https://inkscape.org) and GIMP v2.10.24 (https://www.gimp.org).

**Figure 3 ijms-23-12974-f003:**
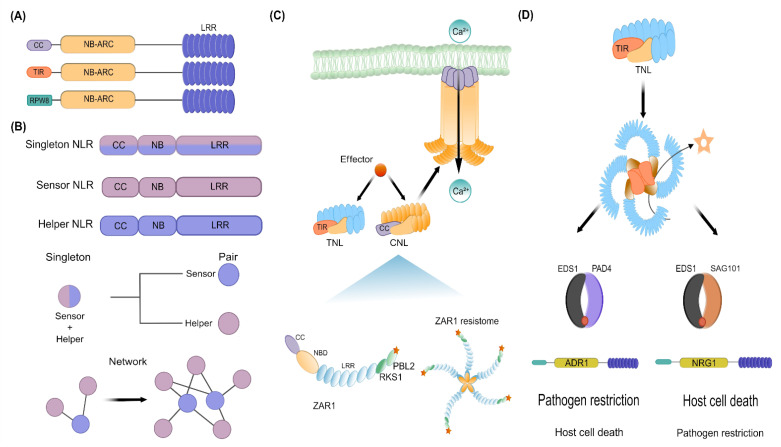
An overview of NLR-mediated ETI. (**A**) Classification of NLRs according to their N-terminal domains: CNLs, TNLs, and RNLs. (**B**) An overview of NLRs functioning as a singleton, pair or part of a network. (**C**) Perception of effectors by CNLs and TNLs is depicted, showing the mechanism of ZAR activation as an example of the CNL-mediated immunity. (**D**) TNL-mediated immunity is mediated by two modules (EDS1-PAD4-ADR1 and EDS1-SAG101-NRG1), deriving in two opposite outputs. See the main text for details. The network depicted in 3B was adapted with permission from [84], whereas the lower section of 3D was adapted with permission from [85]. Created with Inkscape (https://inkscape.org) and GIMP v2.10.24 (https://www.gimp.org).

**Figure 4 ijms-23-12974-f004:**
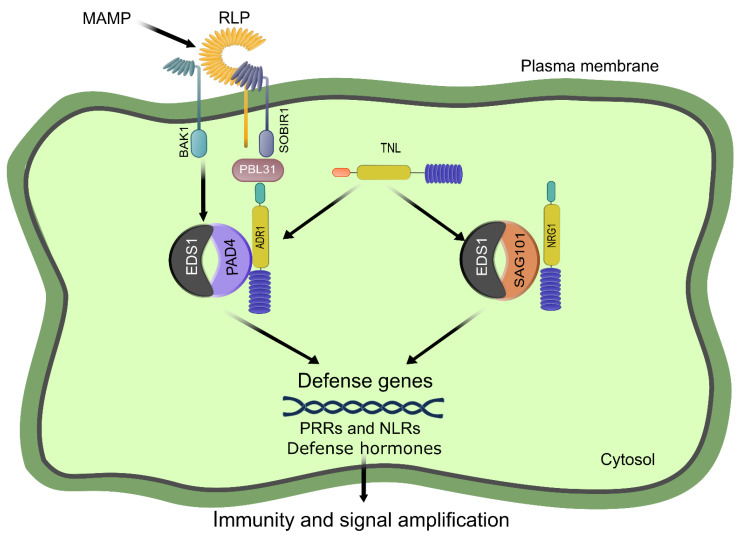
Convergence and integration of PRR- and NLR-mediated immunities. The current evidence connecting PTI and ETI through TNL signaling is shown. Such a point of convergence is the EDS1-PAD4-ADR1 node. See the main text for details. This figure was adapted with permission from [146]. Created with Inkscape (https://inkscape.org) and GIMP v2.10.24 (https://www.gimp.org).

**Figure 5 ijms-23-12974-f005:**
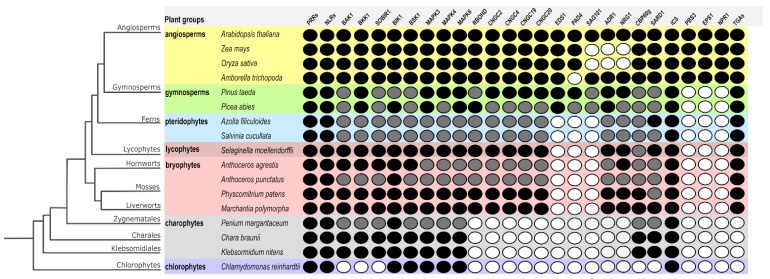
Presence/absence analysis of canonical components connecting PTI and ETI along the plant kingdom. Rows denote species, which are arranged according to [222], with some modifications. Columns denote the presence or absence of known components connecting PTI and ETI. Filled circles indicate presence and clear circles represent absence, as defined by reciprocal BLAST. In the case of filled circles, black-filled denote the result of BLAST plus the domain-oriented orthology inference approach, whereas gray-filled are based on parsimony inference. Created with Inkscape (https://inkscape.org) and GIMP v2.10.24 (https://www.gimp.org).

## Data Availability

Appendix A is publicly available at https://www.mdpi.com/article/10.3390/ijms232112974/s1.

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
