# Peer review of "An Overview of PRR- and NLR-Mediated Immunities: Conserved Signaling Components across the Plant Kingdom That Communicate Both Pathways"

_ijms, 2022, doi:10.3390/ijms232112974_

Round 1
Reviewer 1 Report
The manuscript entitled “An overview of PRR- and NLR-mediated immunities: conserved signaling components across the plant kingdom that communicate both pathways” is devoted to analysis of cell surface-localized pattern recognition receptors (PRRs) and intracellular nucleotide binding domain, and leucine-rich repeat receptors (NLRs) are plant immune proteins that trigger an orchestrated downstream signaling in response to molecules of microbial origin or host plant origin.
However, recent studies reveal that such binary distinction is far from being applicable to the real world.
The review is well-written, and analyze the state-of-art in plant resistance mechanisms linked to PRRs and NLRs.
PRRs are usually associated with pattern-triggered immunity (PTI), whereas NLRs have been involved with effector-triggered immunity (ETI), but such difference seems to be a result of general simplification.
This review is concentrated on analysis of reactions in plants from the point of view of evolutionary conserved pathways across the plant kingdom. Although research on bryophytes is emerging, many genetic traits of other primitive plants have yet to be investigated.
Plants have been engaged in a fight against microbes since their origin. It is speculated that plants evolved two layers of plant immunity, an ancient form of innate immunity known as PTI, and ETI as the second one, which is highly specific.
The Authors have provided that PTI- and ETI-mediated immunities are converging nodes of early signaling mechanisms. PTI and ETI are interdependent in work against microbial infections.
The point of view of the Authors is not original (see Lu Y, Tsuda K. Intimate association of PRR-and NLR-mediated signaling in plant immunity. Molecular Plant-Microbe Interactions. 2021 Jan 29;34(1):3-14.; Rhodes J, Zipfel C, Jones JD, Ngou BP. Concerted actions of PRR-and NLR-mediated immunity. Essays in biochemistry. 2022 Sep;66(5):501-11; Bernoux M, Zetzsche H, Stuttmann J. Connecting the dots between cell surface-and intracellular-triggered immune pathways in plants. Current Opinion in Plant Biology. 2022 Oct 1;69:102276.), but can be found interesting in some aspects.
It will be of high interest for all plant pathologists, microbiologists, and plant scientists. It can be published at present form.
Reviewer 2 Report
This manuscript nicely summarized progresses in understanding plant immune system. It was well organized, and I really like to read this manuscript. It can be accepted after minor revision. I only have one suggestion. There were too many references in the current version. It is suggested to keep the most important ones.
Reviewer 3 Report
A review article addresses a very hot topic in plant physiology: receptor-mediated immunity. The review mentions both well-known facts and provides new information that refines and details the features of signaling events in plant cells associated with the perception and transmission of signals.
The disadvantages of the work include the excessive use of abbreviations for receptor proteins, such as: BAK1/SERK3 and the like, sometimes they can be replaced by the terms "protein kinase" or "proteins". In addition, the following abbreviations are missing in the text: EvoMPMI (p. 703) and RBOHD (p. 400).
Page 38-39. It is not necessary to combine nematodes, protists and bacteria with fungi in one term "microbes".
I recommend the authors to reduce the list of references to 100-120 sources.
Dr. Lomovatskaya L.
